# Linear-Time Algorithms for Representative Subset Selection From Data Streams

## Abstract

Representative subset selection from data streams is a critical problem with wide-ranging applications in web data mining and machine learning, such as social media marketing, big data summarization, and recommendation systems. This problem is often framed as maximizing a monotone submodular function subject to a knapsack constraint, where each data element in the stream has an associated cost, and the goal is to select elements within a budget $B$ to maximize revenue. However, existing algorithms typically rely on restrictive assumptions about the costs of data elements, and their performance bounds heavily depend on the budget $B$. As a result, these algorithms are only effective in limited scenarios and have super-linear time complexity, making them unsuitable for large-scale data streams. In this paper, we introduce the first linear-time streaming algorithms for this problem, without any assumptions on the data stream, while also minimizing memory usage. Specifically, our single-pass streaming algorithm achieves an approximation ratio of $1/8 - \epsilon$ under $O(n)$ time complexity and $O(k \log \frac{1}{\epsilon})$ space complexity, where $k$ is the largest cardinality of any feasible solution. Our multi-pass streaming algorithm improves this to a $(1/2 - \epsilon)$-approximation using only three passes over the stream, with $O(\frac{n}{\epsilon} \log \frac{1}{\epsilon})$ time complexity and $O(\frac{k}{\epsilon} \log \frac{1}{\epsilon})$ space complexity. Extensive experiments across various applications related to web data mining and social media marketing demonstrate the superiority of our algorithms in terms of both effectiveness and efficiency.

## CCS Concepts

• **Information systems** → **Web mining**; • **Theory of computation** → **Streaming, sublinear and near linear time algorithms**; **Approximation algorithms analysis**.

## Keywords

web data mining, streaming algorithm, data summarization, submodular maximization

**ACM Reference Format:**
Anonymous Author(s). 2025. Linear-Time Algorithms for Representative Subset Selection From Data Streams. In *Proceedings of (WWW '25)*. ACM, New York, NY, USA, 12 pages. https://doi.org/XXX.XXX

## 1 Introduction

Representative subset selection from large datasets is a fundamental problem with various data-driven applications related to web data mining and machine learning, including but not limited to social media marketing [36, 40, 43, 61], recommendation systems [16, 17, 22, 58, 64], document summarization [13, 48, 52, 54, 55] and feature selection [4, 10, 41, 67]. Many studies (e.g., [6, 15–17, 26, 40, 64]) have formulated this problem as the task of selecting a subset that maximizes a submodular function. This approach leverages the "diminishing returns" property of submodular functions to quantify the "representativeness" or "utility" of the selected subset. Moreover, constraints such as cardinality or knapsack constraints are typically imposed on the objective submodular function to model real-world limitations [3, 5, 16, 26, 40, 56, 64].

Since the seminal work of Fisher et al. [30], constrained submodular maximization problems have been extensively studied, with a variety of algorithms proposed that achieve good approximation ratios [32, 44, 50, 56, 62, 65]. However, the advent of big data has introduced new challenges, rendering many of these algorithms less practical due to their computational demands. Over the past decades, the exponential growth in data size has placed increasing demands on algorithmic efficiency, leading to substantial research efforts to develop faster submodular maximization algorithms. Early work in this line has achieved nearly linear time complexity of $O_\epsilon(n \log k)$ [7, 23–25, 33, 46], where $O_\epsilon$ hides $\epsilon$ factors, $n$ denotes the size of the ground set and $k$ denotes the maximum cardinality of any feasible solution[1]. More recent work (e.g., [9, 11, 12, 15, 20, 47, 53, 57, 59]) has focused on further reducing runtime, surpassing the nearly linear complexities of previous approaches and proposing clean linear-time algorithms for submodular maximization problems. Since linear time complexity is the minimum required to read all elements of the ground set, it is unlikely that any algorithm can be more efficient without employing parallelization, while still maintaining a reasonable approximation ratio [53]. Moreover, in many domains, data volumes are expanding at a rate exceeding the capacity of computers to store them in main memory [7]. Therefore, many studies such as [5, 6, 15–17, 26, 58, 64] have focused on memory-efficient algorithms and proposed streaming submodular maximization algorithms, which takes a constant number of passes through the ground set while accessing only a small fraction of the data stored in main memory at any given time.

In this paper, we formulate the representative subset selection problem as the problem of monotone submodular maximization subject to a knapsack constraint (abbreviated as the **SMKC** problem). The knapsack constraint is a fundamental constraint that can capture real-world limitations such as budget, time, or size, and thus the SMKC problem has been extensively studied since 1982 [65].

---

[1]The time complexity of submodular maximization algorithms is typically measured by the number of oracle queries to the objective function, as these queries are significantly more time-consuming than other basic operations [1, 2, 29, 46].

Currently, for this problem, existing work [53, 59] has only successfully proposed linear-time algorithms with provable approximation ratios in the offline setting, where all data must be stored in main memory—an impractical requirement for many real-world applications. In the streaming setting, the current best single-pass [39] (resp. multi-pass [34]) streaming algorithm can only achieve the super-linear time complexity of $O(\frac{n}{\epsilon} \log B)$ and the space complexity of $O(\frac{B}{\epsilon} \log B)$ (resp. $O(\frac{k}{\epsilon} \log B)$) with an approximation ratio of $1/3 - \epsilon$ (resp. $1/2 - \epsilon$), where $B$ is the budget for the knapsack constraint. Note that the value of $B$ also influences the complexities of other existing streaming algorithms for the SMKC problem to the same extent, if not more (refer to Table 1). In the worst case, the budget $B$ can be arbitrarily large and grow exponentially with the input size $n$, resulting in a quadratic or worse complexity for these algorithms. More critically, the approximation ratios of existing algorithms are derived under the assumption that the cost of each element is no less than 1. These algorithms suggest using normalization to ensure the assumption holds, thereby supporting their approximation ratios. However, such normalization is impractical for single-pass streaming algorithms as the costs are not known in advance, rendering their performance guarantees perhaps invalid. Meanwhile, this normalization implies that $B$ cannot be normalized to reduce time and space complexity in these algorithms, further compounding the efficiency issues. Therefore, we aim to answer the following questions in this paper:

- Given that the assumptions and performance guarantees of existing streaming algorithms for the SMKC problem may not always hold in practical scenarios, is it possible to design more practical streaming algorithms for the SMKC problem that maintain provable performance guarantees without relying on restrictive assumptions?
- Furthermore, if such algorithms exist, can they achieve linear time complexity while using minimal memory?

## 1.1 Our Contributions

In this paper, we provide confirmative answers to the above questions, by presenting two novel streaming algorithms for the SMKC problem without any assumptions on the data stream. The contributions of our paper can be summarized as follows:

- We propose a single-pass streaming algorithm dubbed OneStream that achieves an approximation ratio of $1/8 - \epsilon$ for the SMKC problem. The time and space complexities of the OneStream algorithm are $O(n)$ and $O(k \log \frac{1}{\epsilon})$, respectively. To our knowledge, OneStream is the *first* streaming algorithm with a provable approximation ratio and linear time complexity for the SMKC problem.
- Based on the OneStream algorithm, we further propose a multi-pass streaming algorithm, dubbed MultiStream, which achieves an approximation ratio of $1/2 - \epsilon$ within three passes over the data stream. This matches the best ratio achieved by existing streaming algorithms for the SMKC problem. However, while existing streaming algorithms require super-linear time complexity, our MultiStream algorithm only has a linear time complexity of $O(\frac{n}{\epsilon} \log \frac{1}{\epsilon})$ under $O(\frac{k}{\epsilon} \log \frac{1}{\epsilon})$ space complexity.

- We conduct extensive experiments using several real-world applications related to the web, including maximum coverage on networks and revenue maximization on networks. The experimental results strongly demonstrate the effectiveness and efficiency of our algorithms.

## 1.2 Challenges and Techniques

To our knowledge, existing streaming submodular maximization algorithms with linear time complexity are limited to handling cardinality [15, 47] or matroid constraints [9, 12, 20], and fail to offer performance guarantees for the knapsack constraint. Additionally, many techniques used in these algorithms are specific to cardinality or matroid constraints and do not easily extend to knapsack constraints. For example, the linear-time streaming algorithms for cardinality constraints rely heavily on the fact that a solution with $k$ elements satisfies the constraint. This property is used (1) to control the number of elements maintained by the algorithm, thereby ensuring that memory consumption stays within an acceptable bound of $O(k)$, and (2) to select the last $k$ elements from the tail of the solution set to form the final feasible solution. However, in the SMKC problem, we lack prior knowledge of the value of $k$, and a solution with $k$ elements may not necessarily satisfy the knapsack constraint. Similarly, the performance guarantees of linear-time streaming algorithms for matroid constraints rely on the exchange property of matroids, a characteristic absent in knapsack constraints.

Moreover, existing streaming algorithms for the SMKC problem rely on guessing an "ideal threshold" to achieve their approximation ratios, and they find this threshold through a canonical geometric search process under the assumption that each element's cost is at least 1. However, their threshold guessing approach needs extra time and memory complexity of $O(\log B)$, resulting in unsatisfactory super-linear time complexity, especially since the budget $B$ can be arbitrarily large and even grow exponentially with the input size $n$ in the worst-case scenario. To address cases where elements' costs are less than 1, existing streaming algorithms suggest using normalization to ensure that each element's cost is at least 1. However, such normalization is impractical for single-pass streaming algorithms as the costs are not known in advance, which invalidates their performance guarantees. Furthermore, this normalization implies that $B$ cannot be normalized to reduce time and space complexity in these algorithms, further compounding the efficiency issues.

To address the above challenges, our OneStream algorithm maintains a "cumulative set" $\bigcup_{t=j}^{i} S_t$, which consists of a small number of candidate solutions $S_t : t \in [j, i]$. Each candidate solution $S_t$ is initialized as an empty set and grows by adding elements from the data stream until it becomes a "nearly feasible solution", i.e., a set that satisfies the knapsack constraint by removing at most one element. OneStream also computes a threshold based on the utility of the cumulative set to control the cost-effectiveness of elements added to the candidate solutions. This approach eliminates the need for the time-consuming geometric search process used in previous streaming algorithms to find an ideal threshold. By constructing nearly feasible solutions, OneStream offers two key benefits: (1) it limits the size of each candidate solution to at most $k + 1 = O(k)$;

**Table 1: Streaming algorithms for monotone submodular maximization subject to a knapsack constraint.**

| Passes | Reference | Ratio | Space Complexity | Time / Query Complexity |
|---|---|---|---|---|
| $= 1$ | [17] | $1/6 - \epsilon$ | $O(\frac{k}{\epsilon} \log B)$ | $O(\frac{nk}{\epsilon} \log B)$ |
| | [39] | $4/11 - \epsilon$ | $O(\frac{B}{\epsilon^4} \log^4 B)$ | $O(\frac{n}{\epsilon^4} \log^4 B)$ |
| | [37] | $\mathbf{2/5 - \epsilon}$ | $O(\frac{B}{\epsilon^4} \log^4 B)$ | $O(\frac{n}{\epsilon^4} \log^4 B)$ |
| | [39] | $1/3 - \epsilon$ | $O(\frac{B}{\epsilon} \log B)$ | $O(\frac{n}{\epsilon} \log B)$ |
| | Algorithm 1 | $1/8 - \epsilon$ | $O(k \log \frac{1}{\epsilon})$ | $O(n)$ |
| $> 1$ | [66] | $1/2 - \epsilon^{\#}$ | $O(B)$ | $O(n(\frac{1}{\epsilon} + \log B))$ |
| | [38] | $\mathbf{1/2 - \epsilon}$ | $O(\frac{B}{\epsilon^7} \log^2 B)$ | $O(\frac{B}{\epsilon^8} \log^2 B)$ |
| | [34] | $\mathbf{1/2 - \epsilon}$ | $O(\frac{k}{\epsilon} \log B)$ | $O(\frac{n}{\epsilon} \log B)$ |
| | Algorithm 2 | $\mathbf{1/2 - \epsilon}$ | $O(\frac{k}{\epsilon} \log \frac{1}{\epsilon})$ | $O(\frac{n}{\epsilon} \log \frac{1}{\epsilon})$ |

$^1$ $k$ denotes the largest cardinality of any feasible solution, $B$ is the budget. Bold font and magenta color indicate the best result(s) in each setting.

$^2$ Apart from our results, the approximation ratios in existing work are based on the assumption that each element's cost is at least 1. They suggest normalization to enforce this assumption, thereby supporting their approximation ratios. However, such normalization is impractical for single-pass streaming algorithms as the costs are not known in advance, potentially invalidating their performance guarantees.

$^3$ In the worst case, $B$ can be arbitrarily large and even grow exponentially with $n$, resulting in a complexity worse than $O(n^2)$ for algorithms dependent on $B$. Note that normalization may not be applicable for reducing $B$, as these algorithms rely on it to ensure the cost assumption and uphold their performance guarantees.

$^{\#}$ The approximation ratio is derived from flawed analysis as pointed out by [34].

and (2) it ensures the final solution (a subset of the cumulative set satisfying budget $B$) is of high quality, as each element in the cumulative set has a cost-effectiveness ratio above the computed threshold. Besides, OneStream employs a sliding window mechanism to control the total number of nearly feasible solutions stored in memory. Thanks to these techniques, OneStream achieves linear time complexity with minimal memory usage.

Our MultiStream algorithm leverages the OneStream algorithm to efficiently guess the "ideal threshold" (associated with the utility of the optimal solution), which guides a better selection of elements to obtain an improved approximation ratio. Different from existing threshold guessing approaches, we dynamically choose an easier-to-find ideal threshold based on the cost distribution of elements in the optimal solution in our proof, which is combined with OneStream's ability to provide accurate upper and lower bounds for the utility of the optimal solution in linear time, enabling an efficient threshold guess process without relying on the assumption that each element's cost is at least 1. This guess process incurs only a small amount of extra $O(1/\epsilon)$ (rather than $O(\log B)$) time and memory overhead, ensuring that MultiStream achieves a better approximation ratio while maintaining linear complexities. More details about our algorithms can be found in Section 4-5.

Due to the space limit, we defer the detailed proofs of most lemmas and theorems to Appendix A, while only providing some intuitions and key ideas for them in the main text.

## 2 Related Work

### 2.1 Algorithms for Monotone Submodular Maximization Under a Knapsack Constraint

Monotone submodular maximization under a knapsack constraint (i.e., the SMKC problem) has been extensively studied [27, 44, 62, 65]

in the offline setting. Among these works, [62] achieved the optimal approximation ratio of $1 - 1/e$, but its $O(n^5)$ time complexity renders it impractical for real-world applications. Subsequent studies [7, 23, 49, 63, 66] put effort into more efficient algorithms, and recent work [53, 59] have proposed linear-time algorithms for the SMKC problem, where [53] achieves $(1/2 - \epsilon)$-approximation using $O(\frac{n}{\epsilon} \log \frac{1}{\epsilon})$ time complexity. However, these algorithms are still limited to the offline setting, requiring all elements to be stored in memory, which is impractical in many real-world scenarios.

Recently, great efforts have been devoted to designing streaming algorithms for the SMKC problem, as shown by Table 1. Among single-pass streaming algorithms, [37] achieves the best approximation ratio of $2/5 - \epsilon$ with time complexity of $O(\frac{n}{\epsilon^4} \log^4 B)$, while [39] offers the best time complexity of $O(\frac{n}{\epsilon} \log B)$ with a worse approximation ratio of $1/3 - \epsilon$. Among multi-pass streaming algorithms, [38] first achieved an approximation ratio of $1/2 - \epsilon$ using $O(\frac{1}{\epsilon})$ passes over the data stream, with time complexity of $O(\frac{B}{\epsilon^8} \log^2 B)$. Subsequently, [66] developed a new streaming algorithm aimed at avoiding the large polynomial factors of $1/\epsilon$ in [38]'s complexity, reducing the time complexity to $O(n(\frac{1}{\epsilon} + \log B))$ while maintaining the same approximation ratio and number of passes over the data stream. However, [34] later pointed out errors in the theoretical analysis of [66], invalidating its approximation guarantee. [34] proposed a new algorithm that restores the $1/2 - \epsilon$ approximation ratio using two passes over the data stream with time complexity of $O(\frac{n}{\epsilon} \log B)$. However, as shown in Table 1, the complexities of all existing streaming algorithms depend on $B$, which, in the worst case, can grow exponentially with the input size $n$, leading to quadratic or worse time complexity for these algorithms. More critically, the approximation ratios of existing algorithms are derived under the assumption that the cost of each element is no less than 1. These

algorithms suggest using normalization to ensure the assumption holds, thereby supporting their approximation ratios. However, such normalization is impractical for single-pass streaming algorithms, rendering their performance guarantees perhaps invalid. Meanwhile, this normalization implies that $B$ cannot be normalized to reduce time and space complexity in these algorithms, further compounding the efficiency issues.

## 2.2 Linear-Time Algorithms for Streaming Submodular Maximization

Chakrabarti and Kale [12] pioneered the achievement of linear complexity for streaming submodular maximization. Their algorithm is tailored for matroid constraints, requires one pass over the data stream, and uses $O(n)$ time complexity and $O(k)$ space complexity to achieve a 1/4 approximation ratio. Subsequently, [20] reduces the number of queries in [12]'s algorithm from $2n$ to $n$, while maintaining the same approximation ratio and complexities. [9] further generalize [20]'s algorithm to handle not necessarily monotone submodular functions, achieving a 1/11.66 approximation ratio with the same complexities.

For simpler cardinality constraints, [47] proposed a linear-time single-pass (resp. multi-pass) streaming submodular maximization algorithm, achieving 1/4 (resp. $1 - 1/e - \epsilon$) approximation ratio with $O(n)$ (resp. $O(n/\epsilon)$) time complexity and $O(k \log k \log(1/\epsilon))$ (resp. $O(k \log k)$) space complexity. [15] extends [47]'s algorithms to handle not necessarily monotone submodular functions, achieving $1/23.313 - \epsilon$ (resp. $0.25 - \epsilon$) approximation ratio using one (resp. $O(1/\epsilon)$) pass(es) over the data stream with unchanged complexities.

However, as explained in Section 1.2, none of these algorithms offer any approximation guarantee for our SMKC problem, and many techniques used in these algorithms are tailored to cardinality or matroid constraints, which do not readily generalize to knapsack constraints. Thus, whether there exists a linear-time streaming algorithm for the SMKC problem remains an open question.

## 3 Problem Statement

We consider the problem of selecting a representative subset of elements from a streaming dataset $\mathcal{N}$ of size $n$, aiming to maximize a non-negative set function $f : 2^{\mathcal{N}} \mapsto \mathbb{R}_{\geq 0}$. For any subset $S \subseteq \mathcal{N}$, $f(S)$ quantifies the utility of $S$, i.e., how well $S$ represents $\mathcal{N}$ according to some objective. In many data summarization problems (e.g., [6, 15, 17, 26, 58, 64]), the utility function $f(\cdot)$ exhibits an intuitive property known as submodularity characterized by diminishing returns. The function with submodularity can be defined as follows:

*Definition 3.1 (Submodular Function).* A set function $f : 2^{\mathcal{N}} \mapsto \mathbb{R}_{\geq 0}$ is submodular if for all $X \subseteq Y \subseteq \mathcal{N}$ and $u \in \mathcal{N} \setminus Y$, it holds that $f(u \mid Y) \leq f(u \mid X)$, where $f(u \mid S) = f(S \cup \{u\}) - f(S)$ represents the marginal gain of $u$ with respect to $S$ for any $S \in \{X, Y\}$.

Intuitively, submodularity implies that adding an element $u$ to a set $Y$ yields no more utility gain than adding $u$ to a subset $X$ of $Y$. Besides, $f(\cdot)$ is monotone if $f(X) \leq f(Y)$ for all $X \subseteq Y \subseteq \mathcal{N}$, indicating that adding a new element never decreases the utility. In this paper, we assume that the utility function $f(\cdot)$ is monotone and submodular. Furthermore, we consider a fundamental constraint

that the feasible solution follows a knapsack constraint, which can model real-world constraints such as budget, time, and size.

Assume that each element $u \in \mathcal{N}$ has an associated cost $c(u)$, and the total cost of a set $S \subseteq \mathcal{N}$ is defined as a modular function $c(S) = \sum_{u \in S} c(u)$. Our subset selection problem can then be formulated as the problem of s̲ubmodular m̲aximization subject to a k̲napsack c̲onstraint (abbreviated as the **SMKC** problem):

$$\max\{f(S) : S \subseteq \mathcal{N} \land c(S) \leq B\},$$

where $f(\cdot)$ is submodular and monotone; $B \geq 0$ is the given budget. Following common practice in submodular optimization, we assume that there exists an oracle that can return the value of $f(S)$ for any $S \subseteq \mathcal{N}$. Oracle queries typically have a significantly higher time complexity than other basic operations, so the efficiency of submodular optimization problems is commonly measured by the number of oracle queries [1, 2, 29, 46]. We study the SMKC problem in the streaming setting, where elements in $\mathcal{N}$ arrive sequentially in an arbitrary order. The streaming algorithm is allowed to make a few passes over the elements, using a small memory.

Without loss of generality, we assume that $c(u) \leq B$ for every $u \in \mathcal{N}$, as any element with a cost exceeding the budget can be immediately discarded upon arrival. Throughout this paper, we denote an optimal solution to the SMKC problem as $O$, the element with the highest cost in $O$ as $o_m$ (i.e., $o_m = \arg\max_{u \in O} c(u)$), and the maximum cardinality of any feasible solution as $k$. For notational convenience, let $[i] = \{1, \ldots, i\}$ for any natural number $i$.

## 4 The Single-Pass Streaming Algorithm

In this section, we propose our single-pass streaming algorithm dubbed OneStream, which is the *first* streaming algorithm with a provable approximation ratio and linear time/query complexity for the SMKC problem. Moreover, OneStream does not rely on the assumption that each element's cost is at least 1, which may not hold in single-pass scenarios where elements cannot be normalized in advance. This makes OneStream more practical than existing single-pass streaming algorithms.

### 4.1 Algorithm Design

As shown by Algorithm 1, the OneStream algorithm maintains a "cumulative set" $\bigcup_{t=j}^{i} S_t$ composed of a small number of candidate solutions, where each candidate solution is initialized as an empty set when first added to the cumulative set and then gradually grows by incorporating valuable elements from the data stream until it becomes a "nearly feasible solution" (a set that satisfies the knapsack constraint after removing no more than one element). More specifically, the algorithm uses a threshold based on the utility (i.e., objective function value) of the cumulative set to check the marginal cost-effectiveness (i.e., marginal gain/cost) of each incoming element from the data stream (Line 4). If the element satisfies the threshold requirement, it is added to the most recent candidate set $S_i$ in the cumulative set (Line 5). Once this candidate set grows as a nearly feasible solution, a new candidate set $S_{i+1}$ is initialized to receive elements that meet the threshold requirement, and the process repeats (Line 6 and Line 9). By using the threshold based on the utility of the cumulative set itself, the OneStream algorithm avoids the geometric search for a suitable threshold, thus achieving

---

**Algorithm 1:** OneStream ($\hbar$)

**Input:** integer $\hbar \geq 1$

1 initialize $i \leftarrow 1$, $j \leftarrow 1$, $S_i \leftarrow \emptyset$ and $e^* \leftarrow$ null;

2 take a new pass over the data stream;

3 **while** there is an incoming element $e$ **do**

4    **if** $\frac{f(e|\bigcup_{t=j}^{i} S_t)}{c(e)} \geq \frac{f(\bigcup_{t=j}^{i} S_t)}{B}$ **then**

5      $S_i \leftarrow S_i \cup \{e\}$;

6      **if** $c(S_i) \geq B$ **then**

7        **if** $i - j + 1 = 2\hbar$ **then**

8          Delete sets $S_j, S_{j+1}, \cdots, S_{j+\hbar-1}$; $j \leftarrow j + \hbar$;

9        $i \leftarrow i + 1$; $S_i \leftarrow \emptyset$;

10    $e^* \leftarrow \arg\max_{u \in \{e^*, e\}} f(\{u\})$;

11 $i_n \leftarrow i$; $j_n \leftarrow j$;

12 **if** $c(\bigcup_{t=j_n}^{i_n} S_t) \leq B$ **then** $Q^* \leftarrow \bigcup_{t=j_n}^{i_n} S_t$ ;

13 **else**

14    let $S(x)$ denote the set of the *last x* elements added to $\bigcup_{t=j_n}^{i_n} S_t$; find $z \in [|\bigcup_{t=j_n}^{i_n} S_t|]$ such that $c(S(z)) \leq B \land c(S(z+1)) > B$;

15    $Q^* \leftarrow \arg\max_{Q \in \{S(z), e^*\}} f(Q)$;

16 **return** $Q^*, \bigcup_{t=j_n}^{i_n} S_t$

---

linear complexity. Moreover, constructing candidate solutions as nearly feasible solutions offers a two-fold benefit: (1) limiting the cardinality of each solution to no more than $k + 1 = O(k)$, ensuring less memory consumption; (2) ensuring the cost of each solution is no less than $B$, coupled with the fact that the cost-effectiveness of each element in the solution surpasses the threshold, guarantees a satisfactory overall utility.

OneStream algorithm also employs a sliding window mechanism to control the number of the nearly feasible solution, to ensure the total memory consumption is small. Specifically, if the total number of these solutions reaches the predefined limit of $2\hbar$, OneStream deletes the oldest $\hbar$ sets from the cumulative set $\bigcup_{t=j}^{i} S_t$ (Line 7-8). Recall that the threshold used to add elements in Line 4 depends on $f(\bigcup_{t=j}^{i} S_t)$, which increases as the algorithm runs. Thus, elements added earlier are tested by lower thresholds and are likely to have lower utility. Consequently, deleting these older elements results in only a small loss in utility.

When the data stream ends, the cumulative set $\bigcup_{t=j}^{i} S_t$ may be an unfeasible solution. To extract a good feasible solution from it as the final output, OneStream searches for a feasible solution $S(z)$ from the tail of $\bigcup_{t=j}^{i} S_t$ (Line 14). The intuition behind this is that elements added later have passed the test by higher thresholds and are more likely to possess good utility. The one with better utility between $S(z)$ and the best singleton element set (generated by Line 10) is then returned as the final solution $Q^*$ (Line 15).

## 4.2 Theoretical Analysis

Our overall analysis approach is as follows. We first demonstrate that the complete "cumulative set" $\bigcup_{t=1}^{i} S_t$ without any deletions can provide an upper bound for the utility of the optimal solution

upon the termination of the algorithm (Lemma 4.1); then show that the utility loss caused by deleting old nearly feasible solutions from the cumulative set is small and can be bounded (Lemma 4.2-4.4). Based on these, we can prove that the final solution obtained after solution deletion and element extraction can also upper bound the utility of the optimal solution, resulting in the approximation ratio of the algorithm (Lemma 4.5).

LEMMA 4.1. *Upon termination of Algorithm 1, the following inequality holds: $f(\bigcup_{t=1}^{i_n} S_t) \geq f(O)/2$.*

To demonstrate that the utility loss caused by deleting nearly feasible solutions in Line 8 is small, we first demonstrate that incorporating a new nearly feasible solution $S_t$ into the cumulative set $\bigcup_{t=j}^{q-1} S_t$ doubles the utility of it (Lemma 4.2), resulting in a continuous increase in the utility of the cumulative set as OneStream runs, even when the deletion occurs (Lemma 4.3).

LEMMA 4.2. *At the end of the each iteration of the **while** loop in Algorithm 1, we must have $f(\bigcup_{t=j}^{q} S_t) \geq 2 \cdot f(\bigcup_{t=j}^{q-1} S_t)$ for any $q \in [j+1, i] : c(S_q) \geq B$.*

LEMMA 4.3. *Let $T_i$ ($i > 1$) denote the state of $\bigcup_{t=j}^{i} S_t$ right before the execution of Line 9 in Algorithm 1. Consider the iteration of the **while** loop in Algorithm 1 when $T_i$ is generated, then:*

- *If the deletion in Line 8 is not executed in the current iteration, we have $2 \cdot f(T_{i-1}) \leq f(T_i)$.*
- *Otherwise, we have $f(T_{i-1}) \leq f(T_i)$.*

Building upon the previous two lemmas, we can demonstrate that solution deletions do not result in a significant utility loss and the cumulative set finally stored in memory retains a substantial utility, as shown by Lemma 4.4.

LEMMA 4.4. $f(\bigcup_{t=1}^{i_n} S_t) \leq (1 + \frac{1}{2^{\hbar-1}-1})f(\bigcup_{t=j_n}^{i_n} S_t)$

Before deriving the approximation ratio of Alg. 1, we only need to prove the solution $Q^*$ returned by the algorithm can upper bound the final cumulative set $\bigcup_{t=j_n}^{i_n} S_t$, based on the observation that elements added later to $\bigcup_{t=j_n}^{i_n} S_t$ have passed higher threshold tests than those added earlier, thus possessing high utility of $\bigcup_{t=j_n}^{i_n} S_t$.

LEMMA 4.5. $f(\bigcup_{t=j_n}^{i_n} S_t) \leq 4 \cdot f(Q^*)$

Combining Lemma 4.1, 4.4 and 4.5, we can immediately get the performance bounds of OneStream, as shown by Theorem 4.6.

THEOREM 4.6. *By setting $\hbar = \log_2(\frac{1}{8\epsilon}) + 1$ where $\epsilon \in (0, 1)$, OneStream can return a solution $Q^*$ satisfying $c(Q^*) \leq B$ and $f(Q^*) \geq (1/8 - \epsilon)f(O)$ for the SMCK problem in a single pass over the data stream. The time/query and space complexities of the algorithm are $O(n)$ and $O(k \log \frac{1}{\epsilon})$, respectively.*

PROOF. The approximation ratio can be directly derived by combining Lemmas 4.1, 4.4, and 4.5. For each incoming element, the algorithm incurs one oracle query at Line 4 and another at Line 10, resulting in a total of $2n$ oracle queries and a time complexity of $O(n)$. As shown in Lines 6-9, the algorithm maintains at most $\hbar$ candidate solutions, each with a size no more than $k + 1$, leading to a space complexity of $O(k \log \frac{1}{\epsilon})$. □

---

**Algorithm 2:** MultiStream $(h, \epsilon)$

**Input:** integer $\hbar \geq 1$ and number $\epsilon \in (0, 1)$

1   $M_1, M_2 \leftarrow$ OneStream$(\hbar)$; $P \leftarrow \{(1 - \epsilon)^{-z} : z \in$

    $\mathbb{Z} \wedge \frac{(1-\epsilon)f(M_1)}{2B} \leq (1-\epsilon)^{-z} \leq \frac{(1+\frac{1}{2^{\hbar-1}-1})f(M_2)}{\epsilon B}\}$;

2   initialize $A_\rho \leftarrow \emptyset$ for each $\rho \in P$ and $L^* \leftarrow M_1$;

3   take a new pass over the data stream;

4   **while** there is an incoming element $e$ **do**

5      **foreach** $\rho \in P \wedge \rho \leq f(e)/c(e)$ **do**

6          **if** $c(A_\rho) + c(e) \leq B \wedge f(e \mid A_\rho) \geq \rho \cdot c(e)$ **then**

7             $A_\rho \leftarrow A_\rho \cup \{e\}$;

8          **if** $f(A_\rho) > f(L^*)$ **then**

9             $L^* \leftarrow A_\rho$;

10   take a new pass over the data stream;

11   **while** there is an incoming element $e$ **do**

12      **foreach** $\rho \in P$ **do**

13          **if** $e \notin A_\rho \wedge c(A_\rho) + c(e) \leq B \wedge f(A_\rho \cup \{e\}) \geq f(L^*)$ **then**

14             $L^* \leftarrow A_\rho \cup \{e\}$;

15   **foreach** $\rho \in P$ **do**

16      **foreach** $e \in A_\rho$ **do**

17          **if** $e \notin L^* \wedge c(L^*) + c(e) \leq B$ **then** $L^* \leftarrow L^* \cup \{e\}$ ;

18   **return** $L^*$

---

## 5   The Multi-Pass Streaming Algorithm

In this section, we propose our multi-pass streaming algorithm dubbed MultiStream, which improves the approximation ratio to $1/2 - \epsilon$, matching the best ratio achieved by existing streaming algorithms for the SMKC problem while maintaining linear time complexity.

### 5.1   Algorithm Design

As shown by Algorithm 2, MultiStream algorithm first efficiently guess the "ideal threshold" related to the utility (i.e., objective function value) of the optimal solution, based on OneStream algorithm (Line 1). It then performs element selection based on each guessed threshold $\rho \in P$ (i.e., potential ideal threshold) within two passes over the data stream (Line 2-14). The specific element selection process is as follows. For each incoming element in the data stream, the algorithm first selects thresholds from $P$ that are smaller than the current element's cost-effectiveness (Line 5), as the candidate solution with a threshold larger than the element's cost-effectiveness would not accept the element, rendering further checking unnecessary. Then, for each selected threshold $\rho$, the algorithm adds the element to $A_\rho$ if the element satisfies both the knapsack constraint and the threshold requirement (Line 6-7). The candidate solution with the best utility is stored in $L^*$ (Line 8-9). Subsequently, the algorithm re-reads the data stream and attempts to insert each element into each existing candidate solution without violating the knapsack constraint, thereby enhancing the candidate solution's utility (Line 10-14). Finally, the algorithm attempts to insert elements stored in memory into the current optimal solution $L^*$,

further improving the utility of the returned solution in practice (Line 15-17).

### 5.2   Theoretical Analysis

As shown by Lemma 5.1, if the element with the highest cost in the optimal solution (i.e., $o_m$) has a large utility, we can directly conclude the algorithm exhibits a favorable approximation ratio.

**LEMMA 5.1.** *If $f(\{o_m\}) \geq f(O)/2$, the solution returned by $L^*$ Algorithm 2 satisfying $c(L^*) \leq B$ and $f(L^*) \geq (1/2 - \epsilon)f(O)$.*

**PROOF.** The lemma follows as $f(M_1) \leq f(L^*)$ and OneStream ensures that $f(M_1) \geq \max_{u \in \mathcal{N}} f(\{u\})$. □

Now focus on the case where the utility of $o_m$ is relatively small. We divide our following analysis into two cases based on whether $o_m$ consumes the majority of the budget of the optimal solution $O$, and then demonstrate that in both cases, the algorithm can find a candidate solution with a corresponding ideal threshold that can upper bound the utility of the optimal solution (Lemma 5.2-5.3). A key purpose of the case-by-case discussion is to quickly find the ideal threshold without relying on the assumption that the cost of any element is no less than 1, which can be better understood through the following example. The $B - c(o_m)$ in the ideal threshold $\rho^*$ of Lemma 5.3 might be very small or even zero. Therefore, we consider the case where it is greater than or equal to $\epsilon B$, ensuring that $\rho^*$ can be quickly found through geometric search.

The proof ideas for Lemma 5.2 can be explained as follows. We first establish the existence of a candidate solution with the ideal threshold $\rho^*$. Then we show that if the cost of this candidate solution is sufficiently large, its utility is also sufficiently large due to the threshold filtering process. Otherwise, the candidate solution retains enough budget to include all elements in the optimal solution except for $o_m$. Thus, elements in $O \setminus \{o_m\}$ that are excluded from this candidate solution must have low marginal cost-effectiveness, which implies excluding these elements causes little utility loss. The proof of Lemma 5.3 follows a similar line of reasoning.

**LEMMA 5.2.** *If $B - c(o_m) < \epsilon B$ and $f(\{o_m\}) < f(O)/2$, then Algorithm 2 can generate a candidate solution $A_{\rho^*}$ satisfying $c(A_{\rho^*}) \leq B$ and $f(A_{\rho^*}) \geq (1/2 - \epsilon)f(O)$, where $\rho^* \in [\frac{(1-\epsilon)f(O)}{2B}, \frac{f(O)}{2B}]$.*

**LEMMA 5.3.** *If $B - c(o_m) \geq \epsilon B$, then Algorithm 2 can generate a candidate solution $A_{\rho^*}$ satisfying one of the following conditions:*

    *(1) $c(A_{\rho^*}) \leq B$ and $f(A_{\rho^*}) \geq (1/2 - \epsilon)f(O)$*

    *(2) $c(A_{\rho^*} \cup \{o_m\}) \leq B$ and $f(A_{\rho^*} \cup \{o_m\}) \geq (1/2 - \epsilon)f(O)$*

*where $\rho^* \in [\frac{(1-\epsilon)f(O)}{2(B-c(o_m))}, \frac{f(O)}{2(B-c(o_m))}]$.*

Based on the two lemmas above, we can readily derive the performance bounds of MultiStream, as shown in Theorem 5.4.

**THEOREM 5.4.** *By setting $\hbar = O(1)$, MultiStream can return a solution $L^*$ satisfying $c(L^*) \leq B$ and $f(L^*) \geq (1/2 - \epsilon)f(O)$ for the SMCK problem within three passes over the data stream. The time/query and space complexities of the algorithm are $O(\frac{n}{\epsilon} \log \frac{1}{\epsilon})$ and $O(\frac{k}{\epsilon} \log \frac{1}{\epsilon})$, respectively.*

**PROOF.** Since Line 11-14 of Algorithm 2 ensure that $f(L^*) \geq f(A_{\rho^*} \cup \{o_m\})$ when the second condition of Lemma 5.3, we also

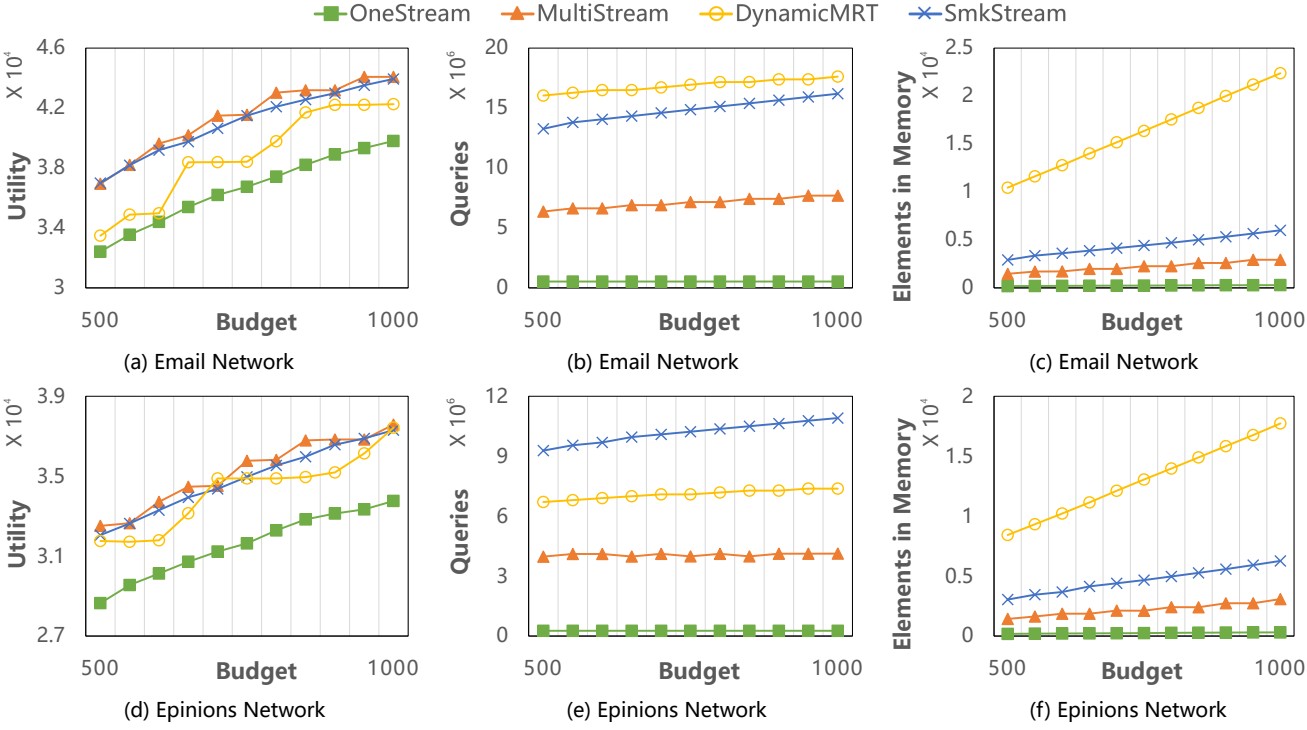

**Figure 1: Experimental results of maximum coverage on networks**

have $f(L^*) \geq f(O)/2$ when the case described by Lemma 5.3 happens. Combining this result with Lemmas 5.1-5.2 yields the approximation ratio.

Note that Algorithm 2 maintains at most $O(|P|)$ candidate solutions. Moreover, for each candidate solution $A_\rho$ ($\rho \in P$), the algorithm incurs at most $n$ oracle queries at Lines 5, 6 and 13, respectively. Thus, the query/time and space complexities of Algorithm 2 are $O(n|P|)$ and $O(k|P|)$, respectively. Based on the definition of $P$ and the fact that $f(M_2) \leq 4 \cdot f(M_1)$ due to Lemma 4.5, we have $|P| = O(\frac{1}{\epsilon} \log \frac{1}{\epsilon})$. Combining these results completes the proof. □

## 6 Performance Evaluation

In this section, we empirically evaluate the performance of our algorithms against the state-of-the-art streaming algorithms for two real-world applications of the SMKC problem, including maximum coverage on networks and revenue maximization on networks. The metrics compared include the utility (i.e., the objective function value), the number of oracle queries to the objective function, and the maximum number of elements in memory. The following four algorithms are implemented in the experiments:

- OneStream: our single-pass streaming algorithm (i.e, Algorithm 1).
- MultiStream: our multi-pass streaming algorithm (i.e, Algorithm 2).
- DynamicMRT [39]: the state-of-the-art single-pass streaming algorithm for the SMKC problem.

- SmkStream [34]: the state-of-the-art multi-pass streaming algorithm for the SMKC problem.

All our experiments are conducted on a Windows workstation with Intel(R) Core(TM) i7-14700 @ 2.10 GHz CPU and 64GB memory[2]. For each of the implemented algorithms, the parameter $\epsilon$ for accuracy (if any) is set to 0.1.

### 6.1 Maximum Coverage on Networks

Maximum coverage has various real-world applications such as web monitoring [60], influence maximization [43, 61], community detection [31] and sensor placement [45]. This application has also been considered in previous studies, such as [5, 16, 17, 19, 21, 26, 64, 68]. Given a network $G = (\mathcal{N}, E)$, our goal is to identify a subset of seed nodes $S \subseteq \mathcal{N}$ that can influence a large number of users within a budget $B$. This goal is formulated as maximizing a monotone submodular function:

$$\max\{f(S) = |\cup_{u \in S} N(u)| : c(S) \leq B\},$$

where $N(u) = \{v : (u, v) \in E\}$ denotes the neighbors of $u$. Following [17, 35, 42], each node $u \in \mathcal{N}$ is associated with a non-negative cost $c(\{u\}) = 1 + \sqrt{d(u)}$, where $d(u)$ represents the out-degree of $u$, and the costs of all nodes are normalized so that the average cost is 2 and the cost of each element is at least 1, ensuring that the approximation ratios of baselines are valid. In our experiments, we use two network datasets sourced from SNAP [51]: (1) the epinions

---

[2]The code is available at: https://anonymous.4open.science/r/LinearKnapStream/

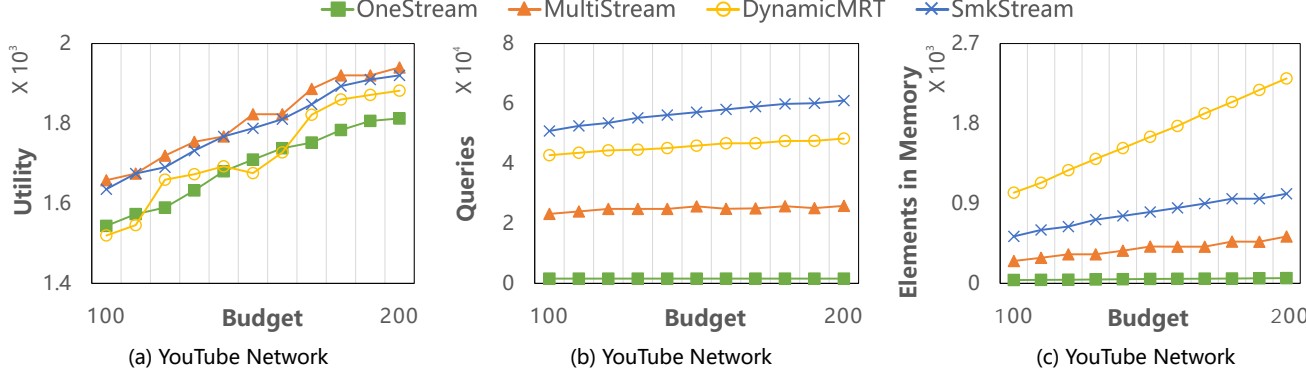

Figure 2: Experimental results of revenue maximization on networks

network with 131,828 nodes and 841,372 edges; and (2) the email network with 265,214 nodes and 420,045 edges.

Figure 1 shows the experimental results of maximum coverage on networks. It can be observed that our OneStream algorithm achieves almost 90% of the best utility results using only $2n$ oracle queries, which is 3.43% (resp. 3.09%) of the query count of the super-linear time complexity algorithm DynamicMRT (resp. SmkStream) algorithm on average. This demonstrates that our algorithm significantly improves the efficiency of solving the SMKC problem while sacrificing little utility. Moreover, our OneStream algorithm exhibits the lowest memory consumption, occupying only 1.72% (resp. 5.45%) of the memory used by DynamicMRT (resp. SmkStream) algorithm on average, highlighting its exceptional memory efficiency. Regarding our MultiStream algorithm, it consistently achieves the best utility while using significantly fewer queries and lower memory consumption compared to the baseline algorithms with super-linear time complexity (i.e., DynamicMRT and SmkStream). While OneStream and MultiStream both have linear time complexity and small space complexity, we observe that MultiStream performs worse than OneStream on query count and memory consumption in experiments due to the additional $\epsilon$ constant term in complexity.

## 6.2 Revenue Maximization on Networks

This application is based on the social network marketing model proposed by [36], and is considered by many previous studies (e.g., [2, 3, 8, 14, 15, 18, 28, 34, 40, 47]). In this application, we are given a network $G = (\mathcal{N}, E)$ where each node $u \in \mathcal{N}$ represents a user with an associated cost $c(u)$, and each edge $(u, v) \in E$ has a weight $w_{u,v}$ denoting the influence of $u$ on $v$. Our goal is to select a subset $S \subseteq \mathcal{N}$ of seed users within a budget $B$ (i.e., $\sum_{u \in S} c(u) \leq B$), and pay $c(u)$ to each seed user $u \in S$ for advertising products to maximize the total revenue. The revenue function is defined as

$$f(S) = \sum_{u \in \mathcal{N}} \sqrt{\sum_{v \in S} w_{v,u}},$$

which is monotone and submodular as indicated by [14, 47]. Following [8, 18, 28, 34], the network $G$ is constructed by randomly selecting 25 communities from the top 5, 000 communities in the

YouTube social network[51]; the edge weights are randomly sampled from the continuous uniform distribution $\mathcal{U}(0, 1)$; the cost of any user $u \in \mathcal{N}$ is determined by $c(u) = \sqrt{\sum_{(u,v) \in E} w_{u,v}}$, and the costs of all nodes are normalized so that the average cost is 2 and the cost of each element is at least 1, ensuring that the approximation ratios of baselines are valid.

Figure 2 shows the experimental results for revenue maximization on networks, which further demonstrates the effectiveness of our proposed algorithms. More specifically, our OneStream algorithm achieves approximately 94% of the best utility results, while using only 3.47% (resp. 2.80%) of the query count of the DynamicMRT (resp. SmkStream) algorithm and occupying only 3.00% (resp. 6.17%) of the memory used by DynamicMRT (resp. SmkStream) algorithm. Our MultiStream algorithm consistently achieves the best utility while using significantly fewer oracle queries and lower memory compared to the baseline algorithms with super-linear time complexity (i.e., DynamicMRT and SmkStream). Again, these results demonstrate the superiority of our algorithms in terms of both time and memory usage.

## 7 Conclusion

In this paper, we study the problem of extracting a representative subset from data streams, formulated as maximizing monotone submodular functions subject to a knapsack constraint. Existing streaming algorithms for this problem only achieve super-linear time complexity depending on the budget, potentially reaching quadratic or even higher complexities in the worst case. Moreover, these algorithms rely on a restrictive assumption which may render their performance guarantees invalid in practical scenarios. To address these limitations, we propose a more practical single-pass streaming algorithm that does not depend on such an assumption, achieving an approximation ratio of $1/8 - \epsilon$ with linear complexity of $O(n)$ and space complexity of $O(k \log \frac{1}{\epsilon})$. Furthermore, we propose a multi-pass algorithm achieving an approximation ratio of $1/2 - \epsilon$, matching the best achievable approximation ratio in streaming settings while maintaining linear time complexity and minimum memory usage. The experiments on real-world applications related to web data mining and machine learning demonstrate the superiority of our algorithms in terms of both effectiveness and efficiency.

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

# A Omitted Proofs

## A.1 Proof of Lemma 4.1

PROOF. For any element $u \in \bigcup_{t=1}^{i_n} S_t$, we use $\mathcal{S}^{<u}$ to denote the state of $\bigcup_{t=j}^{i} S_t$ at the moment when the element $u$ is considered by Line 4 of Algorithm 1. Then we have:

$$f(O) - f(\bigcup_{t=1}^{i_n} S_t) \leq \sum_{u \in O \setminus \bigcup_{t=1}^{i_n} S_t} f(u \mid \bigcup_{t=1}^{i_n} S_t)$$

$$\leq \sum_{u \in O \setminus \bigcup_{t=1}^{i_n} S_t} f(u \mid \mathcal{S}^{<u}) < \sum_{u \in O \setminus \bigcup_{t=1}^{i_n} S_t} \frac{f(\mathcal{S}^{<u})}{B} \cdot c(u)$$

$$\leq \sum_{u \in O \setminus \bigcup_{t=1}^{i_n} S_t} \frac{f(\bigcup_{t=j_n}^{i_n} S_t)}{B} \cdot c(u) \leq f(\bigcup_{t=j_n}^{i_n} S_t),$$

where the first and second inequalities are due to the submodularity of $f(\cdot)$; the third inequality is due to Line 4 of Algorithm 1; the fourth inequality is due to the monotonicity of $f(\cdot)$; the last inequality is due to $c(O) \leq B$. □

## A.2 Proof of Lemma 4.2

PROOF. For any element $u \in \bigcup_{t=1}^{i_n} S_t$, we use $\mathcal{S}^{<u}$ to denote the state of $\bigcup_{t=j}^{i} S_t$ at the moment when the element $u$ was considered by Line 4. Suppose that the elements in $S_q$ are $\{u_1, \cdots, u_{|S_q|}\}$ (listed according to the order they are added into $S_q$), then we have

$$f(\bigcup_{t=j}^{q} S_t) - f(\bigcup_{t=j}^{q-1} S_t) = \sum_{u_x \in S_q} f(u_x \mid \bigcup_{t=j}^{q-1} S_t \cup \{u_1, \cdots, u_{x-1}\})$$

$$\geq \sum_{u_x \in S_q} f(u_x \mid \mathcal{S}^{<u_x}) \geq \sum_{u_x \in S_q} \frac{f(\mathcal{S}^{<u_x})}{B} \cdot c(u_x)$$

$$\geq \frac{f(\bigcup_{t=j}^{q-1} S_t)}{B} \sum_{u_x \in S_q} c(u_x) \geq f(\bigcup_{t=j}^{q-1} S_t),$$

where the first inequality is due to submodularity and the fact that $\bigcup_{t=j}^{q-1} S_t \cup \{u_1, \cdots, u_{x-1}\} \subseteq \mathcal{S}^{<u_x}$; the second inequality is due to Line 4 of Alg. 1; the third inequality is due to the monotonicity. □

## A.3 Proof of Lemma 4.3

PROOF. If the deletion in Line 8 is not executed, the lemma follows directly from $T_i = T_{i-1} \cup S_i$ and Lemma 4.2

Now, consider the case where the deletion in Line 8 is executed. In this case, we have $T_i = T_{i-1} \setminus T_{i-\hbar} \cup S_i$. Thus, we can get

$$f(T_i) = f(T_{i-1} \setminus T_{i-\hbar} \cup S_i) \geq f(T_{i-1} \cup S_i) - f(T_{i-\hbar})$$

$$\geq 2 \cdot f(T_{i-1}) - \frac{f(T_{i-1})}{2^{\hbar-1}} = (2 - \frac{1}{2^{\hbar-1}})f(T_{i-1}) \geq f(T_{i-1}),$$

where the first inequality is due to the submodularity of $f(\cdot)$; the second inequality is due to Lemma 4.2 and the fact that there are $\hbar - 1$ sets are added into $T_{i-1}$ without any deletion from $T_{i-\hbar}$ to $T_{i-1}$; the last inequality is due to the fact that $\hbar \geq 1$.

Combine all the above and finish the proof. □

## A.4 Proof of Lemma 4.4

PROOF. By the submodularity of $f(\cdot)$, we have

$$f(\bigcup_{t=1}^{i_n} S_t) \leq f(\bigcup_{t=1}^{j_n-1} S_t) + f(\bigcup_{t=j_n}^{i_n} S_t) \qquad (1)$$

Note that $\bigcup_{t=1}^{j_n-1} S_t$ can be written as multiple unions of $S_t$ ($t \in [j_n - 1]$), where each union consists of $\hbar$ disjoint $S_t$ that are deleted by Line 8 of Algorithm 1. Thus, we can prove this lemma by showing that the loss in utility caused by these deleted sets can be bounded by the final cumulative set $\bigcup_{t=j_n}^{i_n} S_t$.

Suppose that a total of $M$ deletions occur during the algorithm's execution; denote each union of deleted sets as $D_t$ ($t \in [M]$) and arrange them in such a way that $t_1 < t_2$ implies $D_{t_1}$ is deleted after $D_{t_2}$ (in reverse order of deletion). According to the above definition, we have $\bigcup_{t=1}^{j_n-1} S_t = \{D_t : t \in [M]\}$. It can be observed that each $D_t$ has previously appeared as a cumulative set $\bigcup_{t'=j}^{i} S_t = T_i$ ($i, j \in [i_n]$), as illustrated below:

OBSERVATION 1. *For any $D_t$ ($t \in [M]$), there exists a $T_i$ such that $D_t = T_i$, where $i = (M - t + 1)\hbar$ and $T_i$ is defined in Lemma 4.3.*

PROOF. According to Line 6-8 of Algorithm 1, the first deletion occurs in the iteration where $T_{2\hbar} = \bigcup_{t=1}^{2\hbar} S_t \setminus \bigcup_{t=1}^{\hbar} S_t = \bigcup_{t=\hbar+1}^{2\hbar} S_t$ is generated, with the deleted set being $D_M = T_\hbar = \bigcup_{t=1}^{\hbar} S_t$. The second deletion occurs in the iteration where $T_{3\hbar} = \bigcup_{t=2\hbar+1}^{3\hbar} S_t \setminus \bigcup_{t=\hbar+1}^{2\hbar} S_t = \bigcup_{t=2\hbar+1}^{3\hbar} S_t$ is generated, with the deleted set being $D_{M-1} = T_{2\hbar} = \bigcup_{t=\hbar+1}^{2\hbar} S_t$. Following this rule, we have $D_t = T_{(M-t+1)\hbar}$, which completes the proof. □

Combining this observation with Lemma 4.3, we can get

$$f(D_t) = f(T_{(M-t+1)\hbar}) \geq 2^{\hbar-1} \cdot f(T_{(M-t)\hbar}) = 2^{\hbar-1} \cdot f(D_{t+1})$$

Observe that $D_1 = T_{M \cdot \hbar}$ is the last union of sets to be deleted from memory by Algorithm 1, and this deletion occurs when $T_{M \cdot \hbar + \hbar}$ is generated. Therefore, by Lemma 4.3 and the monotonicity of $f(\cdot)$, we can conclude that

$$f(\bigcup_{t=j_n}^{i_n} S_t) \geq f(T_{M \cdot \hbar + \hbar}) \geq 2^{\hbar-1} \cdot f(T_{M \cdot \hbar}) = 2^{\hbar-1} \cdot f(D_1).$$

Using the above two inequalities, we can get $f(\bigcup_{t=j_n}^{i_n} S_t) \geq 2^{t(\hbar-1)} \cdot f(D_{t'})$ for any $t' \in [M]$. Combining this result with Eqn. (1), we conclude that

$$f(\bigcup_{t=1}^{i_n} S_t) \leq f(\bigcup_{t \in [M]} D_t) + f(\bigcup_{t=j_n}^{i_n} S_t)$$

$$\leq \sum_{t=1}^{M} f(D_t) + f(\bigcup_{t=j_n}^{i_n} S_t) \leq \sum_{t=0}^{M} 2^{-t(\hbar-1)} \cdot f(\bigcup_{t=j_n}^{i_n} S_t)$$

$$\leq f(\bigcup_{t=j_n}^{i_n} S_t) \sum_{t=0}^{\infty} 2^{-t(\hbar-1)} = f(\bigcup_{t=j_n}^{i_n} S_t) \frac{1}{1 - 2^{1-\hbar}},$$

where the last inequality follows from the sum of a geometric series. The lemma then follows by rearranging the inequality. □

## A.5 Proof of Lemma 4.5

Proof. If $c(\bigcup_{t=j_n}^{i_n} S_t) \le B$, then $\bigcup_{t=j_n}^{i_n} S_t = Q^*$ and the lemma trivially holds. Therefore, we consider the case that $c(\bigcup_{t=j_n}^{i_n} S_t) > B$ in the following.

For any element $u \in \bigcup_{t=1}^{i_n} S_t$, we use $\mathcal{S}^{<u}$ to denote the state of $\bigcup_{t=j}^{i} S_t$ at the moment when the element $u$ is considered by Line 4 of Algorithm 1. Suppose that the elements in $S(z+1)$ are $\{u_1, \cdots, u_{|S(z+1)|}\}$ (listed according to the order which they arrive), then we have

$$f(\bigcup_{t=j_n}^{i_n} S_t) - f(\bigcup_{t=j_n}^{i_n} S_t \setminus S(z+1))$$

$$= \sum_{u_x \in S(z+1)} f(u_x \mid \bigcup_{t=j_n}^{i_n} S_t \setminus S(z+1) \cup \{u_1, \cdots, u_{x-1}\})$$

$$\ge \sum_{u_x \in S(z+1)} f(u_x \mid \mathcal{S}^{<u_x}) \ge \sum_{u_x \in S(z+1)} \frac{f(\mathcal{S}^{<u_x})}{B} \cdot c(u_x)$$

$$\ge \sum_{u_x \in S(z+1)} \frac{f(\bigcup_{t=j_n}^{i_n} S_t \setminus S(z+1))}{B} \cdot c(u_x)$$

$$\ge f(\bigcup_{t=j_n}^{i_n} S_t \setminus S(z+1)),$$

where the first inequality is due to submodularity and the fact that $\bigcup_{t=j_n}^{i_n} S_t \setminus S(z+1) \cup \{u_1, \cdots, u_{x-1}\} \subseteq \mathcal{S}^{<u_x}$; the second inequality is due to Line 4 of Algorithm 1; the third inequality is due to the monotonicity of $f(\cdot)$; the last inequality is due to the definition of $S(z+1)$ in Line 14.

Rearranging the inequality above, we obtain $f(\bigcup_{t=j_n}^{i_n} S_t) \ge 2 \cdot f(\bigcup_{t=j_n}^{i_n} S_t \setminus S(z+1)) \ge 2(f(\bigcup_{t=j_n}^{i_n} S_t) - f(S(z+1)))$, which means that $2 \cdot f(S(z+1)) \ge f(\bigcup_{t=j_n}^{i_n} S_t)$. We denote by $e'$ the element first added into $S(z+1)$. Applying the submodularity, we can get

$$2 \cdot f(Q^*) \ge f(S(z)) + f(e^*) \ge f(S(z)) + f(e') \ge f(S(z+1)),$$

where the second inequality is due to Line 10 of Algorithm 1; the last inequality is due to the fact that $S(z+1) \setminus e' = S(z)$. Combining the last two inequalities completes the proof. □

## A.6 Proof of Lemma 5.2

Proof. By Lemma 4.1, Lemma 4.4 and the fact that $M_1$ is a feasible solution, we must have

$$\frac{(1-\epsilon)f(M_1)}{2B} \le \frac{(1-\epsilon)f(O)}{2B} \le \rho^* \le \frac{f(O)}{2B} \le \frac{(1+\frac{1}{2^{\hbar-1}-1})f(M_2)}{\epsilon B}.$$

From this inequality and the definition of $P$ in Line 1, Algorithm 2 must generate the solution $A_{\rho^*}$ in Line 2. We divide our following analysis into two cases whether all elements of $O \setminus \{o_m\}$ can be added to $A_{\rho^*}$ without violating the knapsack constraint when the algorithm terminates.

- $c(A_{\rho^*}) > (1-\epsilon)B$:
  In this case, by Line 4-7 of Algorithm 2, we must have:

$$f(A_{\rho^*}) \ge \rho^* \cdot c(A_{\rho^*}) > (1-\epsilon)B\frac{(1-\epsilon)f(O)}{2B} \ge (1/2-\epsilon)f(O).$$

- $c(A_{\rho^*}) \le (1-\epsilon)B$:
  In this case, we can conclude that any $u \in O \setminus (A_{\rho^*} \cup \{o_m\})$ can be added into $A_{\rho^*}$ without violating the knapsack constraint. Thus, the reason it is not added is that its cost-effectiveness is smaller than the threshold $\rho^*$. So we have

$$f(O \setminus \{o_m\}) - f(A_{\rho^*}) \le \sum_{u \in O \setminus (A_{\rho^*} \cup \{o_m\})} f(u \mid A_{\rho^*})$$

$$\le \sum_{u \in O \setminus (A_{\rho^*} \cup \{o_m\})} \rho^* \cdot c(u) \le \frac{f(O)}{2B}\epsilon B \le \epsilon f(O)/2, \quad (2)$$

where the first inequality is due to the submodularity of $f(\cdot)$; the second inequality is due to Line 6 of Algorithm 2. Recall our assumption that $f(\{o_m\}) < f(O)/2$, which implies $f(O \setminus \{o_m\}) \ge 1/2 \cdot f(O)$ due to the submodularity of $f(\cdot)$. Combining this observation with Eqn. (2) completes the proof.

□

## A.7 Proof of Lemma 5.3

Proof. By Lemma 4.1, Lemma 4.4 and the fact that $M_1$ is a feasible solution, we must have

$$\frac{(1-\epsilon)f(M)}{2B} \le \frac{(1-\epsilon)f(O)}{2(B-c(o_m))} \le \rho^*$$

$$\le \frac{f(O)}{2(B-c(o_m))} \le \frac{(1+\frac{1}{2^{\hbar-1}-1})f(M_2)}{\epsilon B}.$$

From this inequality and the definition of $P$ in Line 1, Algorithm 2 must generate the solution $A_{\rho^*}$ in Line 2. We divide our following analysis into two cases based on whether all elements of $O \setminus \{o_m\}$ can be added into $A_{\rho^*}$ without violating the knapsack constraint when the algorithm ends (i.e., whether $c(A_{\rho^*})$ exceeds $B - c(o_m)$ or not).

- $c(A_{\rho^*}) > B - c(o_m)$:
  In this case, by Line 4-7 of Algorithm 2, we must have:

$$f(A_{\rho^*}) \ge \rho^* \cdot c(A_{\rho^*}) > B - c(o_m)\frac{(1-\epsilon)f(O)}{2(B-c(o_m))} \ge (1/2-\epsilon)f(O).$$

- $c(A_{\rho^*}) \le B - c(o_m)$:
  In this case, we can conclude that any $u \in O \setminus A_{\rho^*}$ can be added into $A_{\rho^*}$ without violating the knapsack constraint. Thus, the reason it is not added is that its cost-effectiveness is smaller than the threshold $\rho^*$. So we have

$$f(O) - f(A_{\rho^*} \cup \{o_m\}) \le \sum_{u \in O \setminus (A_{\rho^*} \cup \{o_m\})} f(u \mid A_{\rho^*})$$

$$\le \sum_{u \in O \setminus (A_{\rho^*} \cup \{o_m\})} \rho^* \cdot c(u)$$

$$\le \frac{f(O)}{2(B-c(o_m))}(B - c(o_m)) = f(O)/2,$$

where the first inequality is due to the submodularity of $f(\cdot)$; the second inequality is due to Line 6 of Algorithm 2. Combining all the above then the lemma follows.

□

