# OpenReview forum: "Linear-Time Algorithms for Representative Subset Selection From Data Streams"
_ACM.org/TheWebConf/2025/Conference — WWW 2025 Poster_

### Official Review · Reviewer_RwQv · 2024-11-01

**Novelty:** 5
**Technical Quality:** 6

**Review:**

This submission tries to study linear time and single/multiple-pass algorithms for representative subset selection from data streams, and two real-world applications of the SMKC problems are tested in your experiments.
Main comments and suggestions
-This draft is easy to explore, the proposed single and multiple pass streaming algorithms are tested in two data sets, whose scale is a bit small, it does not provide the experimental results on large-scale real-world data, which would improve the strength of this work.
-The core idea of this article is interesting, your maximum coverage and revenue on networks have large intersections with clustering, your current baselines are not competitive enough.
-You are encouraged to be motivated this work by real-life examples in production and case studies (e.g., revenue increase in companies, etc) to demonstrate the performance and utility of the proposed methods and augment more practical results to make it more comprehensive.
-This draft has the potential to scale to large-scale scenarios including distributed cases, missing related state-of-the-art you may want to compare: Distributed Clustering of Linear Bandits in Peer to Peer Networks, Fast Distributed Bandits for Online Recommendation Systems.
-Elaborating a discussion on the potential various applications and directions of the proposed algorithm would also be helpful to polish this work.
-Last but not least, the theoretical component of this work currently is not impressive
Overall, it's enjoyable to read this manuscript, although there are some jobs that need improvement to better demonstrate its merits. In short, it would be a pleasure to recommend this manuscript towards acceptance.

**Questions:**

See above.

**Reviewer Confidence:**

4: The reviewer is certain that the evaluation is correct and very familiar with the relevant literature

**Scope:**

4: The work is relevant to the Web and to the track, and is of broad interest to the community

---

### Official Review · Reviewer_AZiR · 2024-12-01

**Novelty:** 4
**Technical Quality:** 5

**Review:**

The paper introduces OneStream and MultiStream, two novel algorithms for solving the Submodular Maximization with Knapsack Constraint (SMKC) problem in streaming settings. These algorithms achieve linear time complexity while removing restrictive assumptions from existing approaches. The experimental results demonstrate both algorithms’ superiority in efficiency and memory usage over state-of-the-art methods. The paper is well-organized and provides sufficient details about the problem formulation, algorithmic techniques, and experimental design. However, parts of the theoretical analysis, particularly the intuition behind key lemmas and their implications, could benefit from more concise summaries or visual aids. The work addresses a critical bottleneck in scalable data stream processing by combining practicality (linear time and space complexity) with strong theoretical guarantees.

Pros:
1. First linear-time streaming algorithms for SMKC without restrictive cost assumptions.
2. OneStream and MultiStream significantly reduce computational and memory costs compared to baselines.
3. Provable approximation ratios of 1/8−ϵ (OneStream) and 1/2−ϵ (MultiStream).

Cons:
1. Limited exploration of practical scenarios: The algorithms are not tested on extremely noisy or highly dynamic data streams, which are common in real-world applications.
2. The theoretical insights are complex and may not be easily understood by a broader audience.
3. The choice of baselines is appropriate but limited; additional comparisons with recent, specialized streaming algorithms could strengthen the results.

**Questions:**

1. How do the proposed algorithms perform on data streams with significant noise or mislabeled data? Could the cumulative set mechanism handle such cases effectively?
2. How sensitive are the algorithms to the choice of thresholds in practice? Are there any heuristics or practical guidelines for setting these thresholds?
3. Do the algorithms adapt well to highly dynamic streams where data distributions change over time? If not, what modifications would be required?
4. Why were specific baselines chosen for comparison, and could additional benchmarks, particularly those tailored to specific applications (e.g., influence maximization), further validate the methods?

**Reviewer Confidence:**

1: The reviewer's evaluation is an educated guess

**Scope:**

2: The connection to the Web is incidental, e.g., use of Web data or API

---

### Official Review · Reviewer_4JNB · 2024-12-01

**Novelty:** 5
**Technical Quality:** 4

**Review:**

This paper tackles the SMKC problem by proposing efficient single-pass and multi-pass streaming algorithms that operate in linear time without relying on restrictive assumptions. Experimental results show the efficiency and effectiveness of the proposed methods in two real-world applications.

Quality: This work demonstrates good technical rigour and methodological soundness.

Clarity: The manuscript is well-written.

Originality: This work contributes new insights to the field.

Significance: This work offers incremental contributions.

Pros：

S1: The algorithms presented in this paper offer a novel approach to solving the SMKC problem without relying on restrictive assumptions, a unique aspect not covered in previous work.

S2: The time and space complexities of the algorithms in the paper are independent of the budget, resulting in relatively good performance and efficiency.

S3: The theoretical analysis provided in the paper is comprehensive and well-developed.

Cons：

W1: The introduction and use of $e^*$ in Algorithm 1 are somewhat confusing. The paper should provide a clearer explanation -- whether it is for the correctness of the algorithm or some other purpose.

W2: The algorithm needs to rely on the assumption that the utility function f is monotone and submodular. How to ensure or test this hypothesis should be discussed in the paper.

W3: In the experiments, $\epsilon$ is set to 0.1 by default, but the choice of $h$ is not explicitly discussed. A detailed investigation into how variations in $h$ and $\epsilon$ affect both the proposed algorithms and the baselines is essential to understanding their performance and robustness under different parameter configurations.

W4: The datasets used in the experiment are not the data stream mainly discussed in this paper, but some graphs. The paper also does not describe how to simulate data streams through these data. This setting is contrary to the purpose of this paper. Please clarify this.

W5: What exactly does “Budget” refer to in the experiment? How to set it? In the first set of experiments, why does the budget start from 500? And in the second experiment, why start with 100?

W6: The paper contains some typographical errors. For example, in Lemma 4.6 and 5.4, SMCK should be corrected to SMKC.

**Questions:**

See W1-W5.

**Reviewer Confidence:**

3: The reviewer is confident but not certain that the evaluation is correct

**Scope:**

3: The work is somewhat relevant to the Web and to the track, and is of narrow interest to a sub-community

---

### Official Review · Reviewer_9ijt · 2024-12-03

**Novelty:** 5
**Technical Quality:** 5

**Review:**

The paper introduces a linear-time algorithm for representative datastream selection in a streaming setting, addressing the limitations of existing approaches that require the entire dataset to be stored in main memory, resulting in high memory consumption. Unlike prior methods, which assume pre-determined costs for normalization, the authors propose both single-pass and multi-pass streaming algorithms. These algorithms leverage a submodular and monotone set extraction function, constrained by a knapsack budget. The paper provides theoretical proofs for key lemmas and extensive experiments, benchmarking the proposed algorithms against a baseline across various real-world web applications.

The paper could be further improved by addressing the following points:

1. The authors assume a monotone submodular function. It would be helpful to clarify whether all real-world applications adhere to this objective function requirement.

2. Before delving into performance evaluation, the authors could elaborate on the performance metrics used and their implications in a streaming context.

3. For revenue maximization on the YouTube network, the state-of-the-art DynamicMRT algorithm demonstrates better utility performance than the OneStream algorithm under certain budget constraints. This discrepancy warrants further explanation.

**Questions:**

It would be helpful if the authors could address the following questions:

1. The proposed algorithm is designed for monotone submodular functions. What percentage of real-world applications satisfy this objective function requirement? Additionally, how would the algorithm perform if the function were non-monotone submodular?

2. The authors set the accuracy parameter to 0.1. Could the authors elaborate on how this value is determined and whether it is specific to the application?

3. For revenue maximization on the YouTube network, the state-of-the-art DynamicMRT algorithm demonstrates better utility performance than the OneStream algorithm under certain budget constraints. This discrepancy requires further explanation.

4. While the number of queries and memory consumption is significantly lower than the baseline, the utility performance shows variability. Could the authors clarify the reasons for this variance?

5. Does the order in which the data stream appears impact the performance of the OneStream or MultiStream solution? A discussion on the effect of element order would be valuable.

**Reviewer Confidence:**

2: The reviewer is willing to defend the evaluation, but it is likely that the reviewer did not understand parts of the paper

**Scope:**

4: The work is relevant to the Web and to the track, and is of broad interest to the community